# Cyberbullying, Aggressiveness, and Emotional Intelligence in Adolescence

**DOI:** 10.3390/ijerph16245079

**Published:** 2019-12-12

**Authors:** María Carmen Martínez-Monteagudo, Beatriz Delgado, José Manuel García-Fernández, Esther Rubio

**Affiliations:** Department of Developmental Psychology and Teaching, University of Alicante, 03690 Alicante, Spain; beatriz.delgado@ua.es (B.D.); josemagf@ua.es (J.M.G.-F.); erc@alu.ua.es (E.R.)

**Keywords:** cyberbullying, aggressiveness, emotional intelligence, adolescence

## Abstract

The devastating consequences of cyberbullying during adolescence justify the relevance of obtaining empirical evidence on the factors that may cause participation in its distinct roles. The goal of this study was to analyze the predictive capacity of aggressiveness (physical aggression, verbal aggression, anger, and hostility) and emotional intelligence (attention, understanding, and emotional regulation) with respect to being a victim, aggressor or victim–aggressor of cyberbullying during adolescence. The Screening for Peer Bullying, the Aggressiveness Questionnaire and the Trait Meta-Mood Scale-24 were administered to a sample of 1102 Spanish secondary education students, aged 12 to 18. In general, results revealed a higher probability of being a victim, aggressor or victim–aggressor as physical aggressiveness and anger increased. On the other hand, results revealed a low probability of being a victim, aggressor or victim–aggressor as emotional understanding and emotional regulation increased. These findings highlight the importance of considering said variables when creating prevention programs to stop or reduce the social and educational issue of cyberbullying during adolescence.

## 1. Introduction

Insults, threats, humiliation, the publishing of confidential information, the violation of privacy, social exclusion, the spreading of rumors, identity theft, dissemination of physical aggressions… Today, numerous adolescents must face these aggressive behaviors, and others, in social networks on a daily basis. The fast and widespread generalization of the use of the information and technology communications (ICTs) has led to an increase in a very common problem in the adolescent population: cyberbullying. Cyberbullying is defined as “An aggressive, intentional act carried out by a group or individual, using electronic forms of contact, repeatedly and over time against a victim who cannot easily defend him or herself” [1] (p. 376). Prevalence rates of cyber-abuse vary significantly in the distinct research studies that have been carried out. Modecki, Minchin, Harbaugh, Guerra, and Runions [2] carried out a meta-analysis, finding prevalence rates that ranged from 4% to 36% for cybervictimization and from 16% to 18% for cyberaggression. These variations are due, in part, to the varying conceptualizations of cyberbullying, the different assessment instruments used, the design and data analysis performed, the cut-off points established to consider the act cyberbullying, etc. However, despite this variability, these studies all reveal a concerning presence of this phenomenon in the adolescent population. 

Similarly, many studies have corroborated the negative consequences on everyone involved in these cyberbullying behaviors, including aggressors and individuals who are not involved (observers) [3,4]. Victims have received the most attention with respect to the consequences of being cyberbullied. Thus, many studies have revealed that cyberbullying victims experience anxiety, depression, stress, fear, low self-esteem, feelings of anger and frustration, helplessness, nervousness, irritability, somatization, sleep disorders, suicidal thinking, and concentrating difficulties that affect school performance [5,6,7,8,9], whereas aggressors are more likely to have moral disconnection, a lack of empathy with victims, problems caused by their aggressive behavior, criminal conduct, consumption of alcohol and drugs, technology dependence, and school absenteeism [10,11,12,13]. Therefore, past empirical studies have tended to focus on examining the results of being a cyberbullying victim or aggressor, in order to intervene and alleviate these consequences. Fewer studies, however, have attempted to identify the potential predictive variables of cyberbullying, so as to effectively prevent and intervene in this social issue. Thus, it is necessary to identify the variables having a protective effect on cyberbullying or the factors or behaviors leading one to bully or be bullied using the ICTs an important aspect when engaging in preventive interventions. Level of aggressiveness is one of the personal variables that has been closely related to peer bullying. Numerous studies have found that anger and hostility are the most relevant predictors of cyberbullying during adolescence [14]. Therefore, past research has shown that aggressiveness is one of the personal characteristics having the closest relationship with school violence, indicating that it is a major predictor of traditional forms of bullying [15,16] and cyberbullying [17,18,19,20,21], especially with respect to aggressors. Aggressive adolescents attempt to demonstrate their power by dominating their classmates through force or intimidation, in this case, using technological resources to do so. In this sense, You and Lim [21], using logistic regression, found associations between high levels of aggressiveness and an increased probability of engaging in cyberbullying. The authors used a sample of 3449 adolescents, aged 12 to 14. Past research, however, had also found that victims of cyberbullying displayed high levels of aggressiveness, antisocial behavior, anger, and hostility [22,23,24,25,26,27,28]. Thus, Hinduja and Patchin [23] confirmed that 30% of the cyberbullying victims experienced symptoms of anger. Along these lines, Kowalski and Limber [12] affirmed that students exposed to bullying over the Internet tend to experience feelings of anger after the bullying. Giménez, Maquillón and Arnáiz [24] revealed higher levels of aggressiveness in adolescent bullies and victims of cyberbullying as compared to those that were not involved in this behavior. Aricak and Ozbay [22], with an extensive sample of 1257 students (aged 13 to 19), found that students with high levels of anger had a greater probability of being a perpetrator or victim of cyberbullying. Therefore, studies have been carried out on adolescents, suggesting that aggressiveness increases the risk of victimization, affirming that victims are more likely to display violent behavior than nonvictims [28]. However, studies differentiating between victims and victims/aggressors affirm that the latter tend to be more aggressive as compared to pure victims [29,30]. Hence, although most of the victims are characterized by their submissive and passive behavior in the face of their aggressor (pure victims), some victims display aggressive and hostile behavior and are therefore referred to as aggressive victims. This suggests that being subject to victimization may damage the student’s empathetic capacity, whereby, instead of understanding the aggressor’s behavior, the victim attempts to protect themselves by engaging in equally hostile behavior [31].

On the other hand, several investigations have analyzed the differences in sex and age (or grade), taking into account the role played in cyberbullying and its aggressiveness, especially with regard to aggressors. It has been concluded that boys harass more in direct cybernetic behaviors (harassment, persecution, and dissemination of degrading images for the victim [32,33]), which coincides with results found regarding bullying, where boys participate more in direct and physical aggressions and girls in indirect and relational aggressions [34]. However, some research has not noticed differences based on sex [35,36]. Ang, Huan, and Florell [17] concluded that proactive aggression also significantly accounted for variance in cyberbullying above and beyond reactive aggression for both boys and girls. Giménez, Maquillón, and Arnáiz [24], using a sample of 1914 adolescents, found no statistically significant differences in the level of aggressiveness of cyber-aggressor boys and girls. The inconsistency in the results obtained by the studies suggests, therefore, the need for more extensive research in this regard, in order to define the role of sex in violence through ICT. With respect to age, various investigations with participants from 11 to 18 years conclude that as age increases, the number of aggressors increases [12,36], and the level of aggression [20,37]. Other studies conclude that physical aggression decreases with age, while verbal aggression remains constant [38] (Olweus, 1994). These differences with respect to sex and age (or grade) in the different cyberbullying roles could make one suspect that the predictive capacity of the different types of aggressiveness could be different depending on the sex or age of the students; however, few studies have analyzed whether the sex or grade would influence the predictive ability of aggressiveness over the role of victim, aggressor or victim–aggressor.

Emotional intelligence (EI) is one of the personal variables that has received the most attention with respect to cyberbullying. Mayer and Salovey [39] define emotional intelligence as the ability to accurately perceive, appraise, and express emotion; to access and generate feelings when they facilitate thought; to understand emotions and emotional knowledge; and to regulate emotions and promote emotional and intellectual growth. Trait Meta-Mood Scale-24 (TMMS-24) [40] is the most widely used self-report measure in Spain and Latin America to assess individual differences in EI [41], and it is based on the ability EI model (see Barchard, Brackett, and Mestre [42] for a review). The characteristics of EI make it one of the most powerful protective factors against the appearance of school bullying or cyberbullying [43,44]. Low levels of emotional intelligence have been found in both the aggressors and victims of cyberbullying. Thus, Baroncelli and Ciucci [45] corroborated that bullying and cyberbullying were related to difficulties in regulating one’s emotions. Casas, Ortega-Ruiz, and del Rey [46] found that victims paid more attention to emotions but had lower levels of emotional clarity and repair, whereas aggressors had lower levels of emotional attention, clarity, and repair. Garaigordobil [47] found a relationship between engaging in cyberbullying and lower emotional attention, clarity, and repair, whereas cyberbullying victimization was related only to an increased emotional attention. Eden, Heiman, and Olenik-Shemesh [48] found that emotional control and management predicted abuse between peers and cyberbullying victimization. Distinct studies have also suggested a low empathy level in cyberbullies towards their victims, as well as low levels of emotional intelligence. Zych, Ttofi, and Farrington [49] found that aggressors revealed lower levels of empathy as compared to those who were not involved in bullying, whereas victims had the same level of empathy as nonvictims. Similarly, it was found that the victim–aggressor group had the lowest level of empathy. Along these lines, Zych, Beltrán, Ortega-Ruíz, and Llorent [50] found that aggressors and victims–aggressors of bullying and cyberbullying had lower levels of social and emotional skills as compared to those that were not involved in this behavior, while victims of bullying and cyberbullying had similar social and emotional skills to those of uninvolved students. Thus, the study of emotional skills is seen as a promising approach that may potentially clarify its implication in distinct roles of cyberbullying.

On the other hand, with respect to sex, most research finds that high levels of EI are negatively and significantly related to cybervictimization in both boys and girls [51]. Rey, Quintana, Mérida, and Extremera [52] found that deficits in EI and its dimensions were positively associated with cybervictimization in boys and girls, but more in women. For women, the deficit in emotional regulation was significantly associated with greater cybervictimization than in the case of men; thus, low emotional regulation predicted greater cybervictimization in girls than in boys. Some authors have shown that women normally report a greater tendency to attention and emotional regulation compared to men [53,54]. Thus, although multiple investigations have corroborated the fundamental role played by EI in cyberbullying, there are few studies that have analyzed the predictive capacity of EI on the different roles of cyberbullying based on sex and age (or grade).

When considering the serious repercussions of cyberbullying for everyone involved, it is necessary and relevant to conduct predictive studies to identify which variables are more likely to lead an individual to become a victim, aggressor or victim–aggressor of cyberbullying in order to establish effective prevention and intervention programs to alleviate this social and educational problem. Therefore, the objective of this study is to assess the predictive capacity of aggressiveness and EI on being a victim, aggressor or victim–aggressor of cyberbullying during adolescence in the total sample. Likewise, it is intended to verify whether the sex or the academic grade of the students acts on the predictive capacity of the aggressiveness and the emotional intelligence on the role of victim, aggressor or victim–aggressor. Based on past studies, it is expected that aggressiveness will be a predictive factor of being a victim, aggressor or victim–aggressor of cyberbullying in the total sample and attending to the sex and grade of the students (Hypothesis 1). Similarly, it is anticipated that EI will act as a predictive variable of being a victim, aggressor, and victim–aggressor of cyberbullying in the total sample and attending to the sex and grade of the students (Hypothesis 2).

## 2. Materials and Methods

### 2.1. Participants

A total of 1102 Spanish high school (E.S.O., based on its initials in Spanish) and baccalaureate students participated in the study (499 boys and 603 girls) aged between 12 and 18 (*M* = 14.30; *DT* = 1.71). Student distribution based on gender and academic grade was as follows: 224 in the 1st grade of E.S.O. (107 boys and 117 girls); 197 in the 2nd grade of E.S.O. (82 boys and 115 girls); 172 in the 3rd grade of E.S.O. (82 boys and 90 girls); 187 in 4th grade of the E.S.O. (80 boys and 107 girls); 204 in the 1st year of baccalaureate studies (94 boys and 110 girls) and 118 in the 2nd year of baccalaureate studies (54 boys and 64 girls). Using the Chi-Square Test for Homogeneity of frequency distribution, it was found that no statistically significant differences existed between the groups of Gender x Course Year (χ^2^ = 6.91; *p* = 0.328).

### 2.2. Measures

Cyberbullying. Screening for Peer Bullying [55].

The Screening for Peer Bullying is a standardized instrument that permits the assessment of bullying and cyberbullying behaviors. In this study, only the cyberbullying subscale was used. This scale assesses 15 electronic bullying behaviors (sending offensive and insulting messages, making offensive calls, spreading photos or videos in YouTube, making anonymous calls to frighten, threaten or bribe), allowing for the identification of victims, perpetrators, and observers of cyberbullying. The questionnaire contains 45 questions to be responded to on a 4-point Likert scale, ranging from 0 (never) to 3 (always). The response system is triangular, since the assessed individual should identify if they have suffered the 15 bullying behaviors as a victim, if they have carried them out as a perpetrator or if they have witnessed these behaviors being carried out by others or have been aware of their occurrence (observer) during the past year. The psychometric studies carried out in the original study support the test’s internal consistency (α > 0.82) [55]. Similarly, the internal consistency rates of the subscales in this study were found to be suitable: cybervictimization (α = 0.95), cyberbullying (α = 0.96), and observation (α = 0.98).

Aggression Questionnaire (AQ) [56]; adapted by Andreu, Peña and Grana [57].

Aggressiveness was assessed using a Spanish adaptation of the Aggression Questionnaire [57]. The questionnaire consists of 29 items, answered using a 5-point Likert scale (1 = Completely false; 5 = Completely truthful). Items were grouped together in four factors: Physical Aggression (manifested with strikes, pushing, and other forms of physical aggression using one’s own body or another object to inflict injury); Verbal Aggression (manifesting with insults, threats, mocking, etc.); Anger (feeling of anger appearing as a result of previous hostile attitudes); and Hostility (attitude involving disgust and cognitive assessment towards others). The Spanish adaptation of the scale has revealed adequate reliability and validity rates [57]. The reliability coefficients of the AQ scores in this study were found to be adequate: Physical Aggression (α = 0.77); Verbal Aggression (α = 0.68); Anger (α = 0.69); Hostility (α = 0.75) and overall score AQ (α = 0.90).

Trait Meta-Mood Scale-24 (TMMS-24) [41]

EI was assessed using the Spanish adaptation of the TMMS-48 scale [58]. The Spanish scale, created by Fernández-Berrocal et al. [41], assessed EI based on 24 items about which the student was to respond on a 5-point Likert-like scale (1 = Not at all in agreement; 5 = In complete agreement). The items consisted of three factors: Emotional Attention (level of attention that the student pays to their emotions), Emotional Understanding (ability to understand, identify and label their emotional states), and Emotional Repair (ability to regulate emotions). Reliability rates of the Spanish adaptation were found to be adequate (Emotional Attention, α = 0.84; Emotional Understanding, α = 0.82; Emotional Repair, α = 0.81). In this study, reliability (α) was 0.88 for Emotional Attention, 0.90 for Emotional Understanding, and 0.88 for Emotional Repair.

### 2.3. Procedure

Individualized interviews were conducted with the directors of the participating schools in order to explain the objective of the study, describe the assessment instruments, request permission, and promote their collaboration. Subsequently, permission was requested from the Department of Education of the Valencia regional government. Consent to carry out the study was also granted by the Ethics Committee of the University of Alicante. At the same time, an informative letter was sent to the parents of the students, requesting that they provide their written consent to their children’s participation. Students completed the questionnaires anonymously during a class period lasting approximately 45 min. Therefore, participation was voluntary, anonymous, and with prior parental consent. The researchers were present during the test administration in order to clarify any potential doubts. Mean administration times for the three tests were: 15 min (Cyberbullying. Screening for Peer Bullying), 10 min (AQ), and 10 min (TMMS-24). Standards for research on human subjects were respected in accordance with the ethical principles of the Declaration of Helsinki.

### 2.4. Data Analysis

To examine the predictive or classificatory capacity of aggressiveness and emotional intelligence on cyberbullying, a binary logistic regression analysis was conducted following the forward stepwise regression procedure based on the Wald test. Logistic modeling allowed for the estimation of an event, occurrence or result’s probability of taking place (e.g., probability of being a victim of cyberbullying) in the presence of one or more predictors (e.g., aggression). This probability is estimated using the so-called odd ratio statistic (OR). If the OR is higher than one, it indicates that the increase of the independent variable leads to an increase in the probability of the occurrence of the event. On the other hand, if the OR is less than one, an increase in the independent variable leads to a decrease in the likelihood of the event taking place. The variables (victims and aggressor) were dichotomized as a function of percentiles 25 and 75, with the objective of identifying the low or high presence of the construct. The group of aggressors–victims was established based on the sum of cybervictimization and cyberaggression scores, to subsequently dichotomize said variable based on the same criteria of the 25 and 75 quantiles. The proportion of cases correctly classified by the logistic models calculated ranged between 70.7% (aggressiveness) and 71.7% (EI) in the sample of victims, between 55% (aggressiveness) and 76.46% (EI) in the sample of aggressors, and between 66% (aggressiveness) and 64.4% (EI) in the sample of victims–aggressors.

## 3. Results

### 3.1. Prediction of Being a Victim, Aggressor or Victim–Aggressor of Cyberbullying with Respect to Aggressiveness

Regarding the prediction of being a victim, aggressor or victim–aggressor of cyberbullying with respect to aggressiveness in the total sample, the OR ratio indicates that the probability of being a victim of cyberbullying increases by 3% and 6% for each point increase in the Physical Aggression and Anger scale, respectively. Equally, regarding the prediction of being a cyberbullying aggressor, the OR of the logistic model indicates that students have a 5% and 8% higher probability of being a cyberbullying aggressor for each point increase in the Physical Aggression and Anger scale, respectively. On the other hand, as regards the prediction of being a victim–aggressor of cyberbullying, the OR ratio indicates that students are 12% more likely to be a victim–aggressor of cyberbullying with every one-unit increase in the Physical Aggression scale and have a 7% greater probability of being a cyberbullying victim–aggressor with every one-unit increase in the Anger scale (see Table 1).

As for the prediction of being a victim, aggressor or victim–aggressor of cyberbullying with respect to aggressiveness in boys, the OR ratio indicates that the probability of being an aggressor of cyberbullying increases by 6% and 3% for each point increase in the Verbal Aggression and Anger scale, respectively. In girls, the OR ratio indicates that: (a) the probability of being a victim of cyberbullying increases by 7% and 4% for each point increase in the Physical Aggression and Verbal Aggression, respectively; (b) the probability of being an aggressor increases by 2% for each point increase in Anger scale; and (c) the probability of being a victim–aggressor increases by 7% and 3% for each point increase in Physical Aggression and Anger scale, respectively.

Regarding the prediction of being a victim, aggressor or victim–aggressor of cyberbullying with respect to aggressiveness in E.S.O students (12–16 years old), the OR ratio indicates that the probability of being an aggressor of cyberbullying increases by 4% and 6% for each point increase in the Physical Aggression and Anger scale, respectively. Equally, in Baccalaureate students (16–18 years old), the OR ratio indicates that the probability of being an aggressor of cyberbullying increases by 7% and 2% for each point increase in the Physical Aggression and Anger scale, respectively.

### 3.2. Prediction of Being a Victim, Aggressor or Victim–Aggressor of Cyberbullying with Regard to EI

In the total sample, the OR showed that the probability of being a victim of cyberbullying decreased by 6% and 3% per unit increase in the Emotional Understanding and Emotional Repair scale, respectively. With regard to the prediction of being an aggressor of cyberbullying, the OR indicates that the probability of being an aggressor of cyberbullying decreased by 6% and 5% for each point increase in the Emotional Understanding and Emotional Repair scale, respectively. Finally, with respect to the prediction of being a victim–aggressor of cyberbullying, the OR of the logistic model showed that the probability decreased by 11% and 10% per unit increase in the Emotional Understanding and Emotional Repair scale, respectively (see Table 2).

In boys, the OR showed that the probability of being a victim of cyberbullying decreased by 5% and 4% per unit increase in the Emotional Understanding and Emotional Repair scale, respectively. The probability of being an aggressor of cyberbullying decreased by 4% and 5% per unit increase in the Emotional Understanding and Emotional Repair scale, respectively. Finally, the OR showed that the probability of being a victim–aggressor of cyberbullying decreased by 7% per unit increase in the Emotional Understanding. In girls, the OR ratio indicates that: (a) the probability of being a victim of cyberbullying decreased by 6% and 4% for each point increase in the Emotional Understanding and Emotional Repair scale, respectively; (b) the probability of being an aggressor decreased by 2%, 5%, and 4% for each point increase in the Emotional Attention, Emotional Understanding, and Emotional Repair scale, respectively; and (c) the probability of being a victim–aggressor of cyberbullying decreased by 5% and 4% for each point increase in the Emotional Understanding and Emotional Repair scale, respectively.

Regarding the prediction of being a victim, aggressor or victim–aggressor of cyberbullying with respect to emotional intelligence in E.S.O students (12–16 years old), the OR ratio indicates that: (a) the probability of being an aggressor of cyberbullying increased by 4% and 6% for each point decrease in the Emotional Understanding and Emotional Repair scale, respectively; (b) the probability of being a victim–aggressor increased by 3% for each point increase in the Emotional Attention and decreased by 5% and 4% for each point increase in the Emotional Understanding and Emotional Repair scale, respectively. In Baccalaureate students (16–18 years old), the OR ratio indicates that: (a) the probability of being a victim of cyberbullying decreased by 8% for each point increase in the Emotional Understanding scale; (b) the probability of being an aggressor decreased by 9% for each point increase in the Emotional Understanding scale; and (c) the probability of being a victim–aggressor of cyberbullying decreased by 7% for each point increase in the Emotional Understanding.

## 4. Discussion

The goal of the present study was to verify the predictive capacity of certain personal characteristics (aggressiveness and EI) and the participation in distinct cyberbullying roles (victims, aggressors, and victims–aggressors). Further, whether the sex or the academic grade of the students acts on the predictive capacity of the aggressiveness and the emotional intelligence on the role of victim, aggressor or victim–aggressor was analyzed. The analysis of the predictive capacity of aggressiveness and EI on the distinct cyberbullying roles offers some valuable information for preventive interventions based on students’ characteristics. Thus, on the one hand, this study has verified that certain factors of aggressiveness serve as predictor variables of cyberbullying, both in victims as well as in aggressors and in the victim–aggressor group. Therefore, in the total sample, the data confirm an increased probability of being either a victim, aggressor or victim–aggressor as the physical aggressiveness and anger increase, thus maintaining hypothesis 1 of this study. The data corroborate past studies that suggest that student aggressiveness may be a risk factor for being an aggressor [17,18,19,20,21], victim [22,23,24,25,26,27,28] or aggressive victim of cyberbullying [29,30]. Garaigordobil [47], using a sample of 3026 students aged between 12 and 18, found that adolescents with high antisocial behavior scores were significantly more involved in situations of bullying and cyberbullying in all of their roles (victims, aggressors, aggressive victims, and observers) and used more aggressive strategies as interpersonal conflict resolution techniques. Unfortunately, most of the studies have assessed aggressiveness as a unitary construct, without considering the diverse facets of aggressiveness (physical, verbal, anger, hostility). This study has demonstrated how physical aggressiveness and anger stand out as predictive factors of cyberbullying in all of its potential roles. In the case of the cyberbullies, the results corroborate past empirical studies [17,18,19,20,21]. It seems reasonable that those students with greater physical aggressiveness and anger levels would be more susceptible to attempting to cause harm to others, using all available means (including the Internet). These data are in accordance with other studies that have revealed the strong relationship between being a traditional bully and maintaining this behavior through the Internet [59,60], so that the aggressive behavior in everyday life may be transferred to virtual environments. On the other hand, this study has also corroborated past works that indicate that aggressiveness is a predictive factor of being a victim [22,23,24,25,26,27,28] or a victim–aggressor of cyberbullying [3,30]. These studies suggest that aggressive or antisocial behavior increases the risk of being victimized. Inappropriate or aggressive social behavior, also via social networks, may lead to a lower peer acceptance, resulting in an increased probability of being victimized. As for the victim–aggressor group, the data also corroborated the idea that physical aggressiveness and anger increase the probability of belonging to this group of students, with the probability being greater than in the group of pure victims. These data are in line with those from studies that suggest that this is the most aggressive student group, and not the pure victims [29,30]. Thus, having a higher level of physical aggressiveness and anger increases the possibility of the student experiencing an attack, in an attempt to develop a social reputation that is characterized by an anticonformist and antisocial position, presenting themselves as a strong individual that is likely to exact revenge in a violent manner in the case of a repeated attack [61]. To ensure this, the student is involved in violent behavior, attacking others via digital media, thereby becoming a victim–aggressor.

With regard to students’ sex, the results slightly differ from those obtained in the total sample. Thus, it has been proven in boys that only physical aggressiveness and anger predict being a cyberbullying aggressor, which implies that none of the aggressive factors were predictors of the role of victims or victims–aggressors in the sample of boys. In girls, physical aggressiveness and verbal aggressiveness act as predictive variables of being victims, anger as a predictive variable of being an aggressor, and physical aggressiveness and anger as predictive variables of being a victim–aggressor. With respect to the grade, in the E.S.O and Baccalaureate sample, physical aggressiveness and anger were predictive variables of being a cyberbullying aggressor; however, aggressiveness does not predict being a victim or victim–aggressor in these groups. The data slightly differ with respect to the aggressiveness variables that are predictive of cyberbullying roles. It can be observed in some cases how aggressiveness only predicts some roles and not others. These results highlight the importance of attending to the sex and grade of the students with the objective of establishing preventive and cyberbullying intervention programs adjusted to the target population.

On the other hand, the EI factors—specifically, those referring to understanding and emotional regulation—have been found to be predictive variables for participation as a victim, aggressor, and victim–aggressor of cyberbullying, thereby supporting hypothesis 2 of this study. The data indicate that having a greater level of understanding and emotional regulation decreases the probability of participating in any of the roles of cyberbullying. The data are consistent with those from studies that have reported low levels of EI in victims, aggressors, and victims–aggressors of cyberbullying [45,46,47,48,49,50]. As for aggressors, it has been widely verified that they lack emotional skills [45,46,49,50], as well as having a low capacity to attend to, understand, and regulate emotions, leading to emotional and psychosocial imbalances, worsening their relationship with peers. However, students with a high ability to understand and regulate their emotions reveal high levels of nonparticipation in cyberbullying behavior, since this emotional understanding and regulation leads to an increased level of empathy towards their victims, thereby decreasing the probability of their being involved in aggressive behavior that would potentially harm their peers [62]. As for the victims and aggressive victim groups, here, the data have corroborated the protective component of emotional understanding and regulation with regard to cyberbullying. A student with a high level of emotional understanding and regulation has the necessary resources to respond to distinct and potentially conflictive situations with their peers, thereby reducing their probability of becoming a cyberbullying victim or victim–aggressor. Thus, for example, in the case of an attack (insults, mocking, humiliation, etc.), students who can understand and regulate their emotions may use these skills to modify their emotions and successfully handle these situations by acting assertively or simply ignoring these attacks. With respect to the sex and grade of the students, the results are similar to those found in the total sample, so that emotional understanding and emotional regulation act in general as predictive variables of the different cyberbullying roles. In some cases, only the emotional understanding variable acts as a predictor (Baccalaureate sample) of the three cyberbullying roles. These data are in accordance with those investigations that have prioritized the relevant role that emotional understanding and regulation plays on cyberbullying [45,46,47,48,49,50].

Finally, this study has certain limitations. First, the causality of the relationships is limited, given the study’s cross-sectional nature. Therefore, it is recommended that future investigations analyze these variables using a longitudinal design. Furthermore, the assessment of the variables through only self-reporting measures may lead to biases, so other methods should also be included, such as peer assessments, evaluations by professors or observational methods. This may be corrected for using additional variable assessment methods. On the other hand, multiple studies have corroborated the relationship between EI and aggressiveness [63,64,65,66]. Disruptive behaviors are related to an emotional deficit, that is, a person with less EI will be more easily involved in the participation of aggressive and antisocial behaviors [66]. Thus, the fact that students with low emotional intelligence have more difficulties in dealing with social situations because they are not able to properly manage their emotions can lead them to act aggressively in situations of uncertainty. In this study, the predictive capacity of these variables was analyzed independently, with the aim of checking the predictive power of each variable alone on the different cyberbullying roles. Future studies could verify whether the relationship between these variables substantially modifies their role with respect to participation in the different roles of cyberbullying. Despite these limitations, this study presents some very relevant results on cyberbullying in terms of its understanding and potential intervention, since it highlights the predictive nature of certain personal characteristics (aggressiveness and EI) in involvement as an aggressor, victim or victim–aggressor. Thus, it reveals the predictive nature of physical aggressiveness and anger in the involvement in the distinct roles and EI, specifically, emotional understanding and regulation, as protective factors for cyberbullying. The results of this study offer scientific evidence that permits the development of specific educational and social prevention and intervention programs that consider these variables, so as to ultimately alleviate this social problem that has such devastating consequences on those involved. Preventive programs may focus on detecting students having higher levels of physical aggressiveness and anger, so as to appropriately reduce or channel this problematic behavior or to help to prevent the development of aggressive behavior. Intervention programs may also include the development of emotional skills, such as those related to emotional understanding and regulation. In addition to specific intervention programs, educational programs may promote and strengthen assertive behavior in response to violent acts and help students to transversally develop emotional and social skills. An increased ability to understand and regulate one’s own emotions and those of their peers may increase the student’s empathetic ability, thereby decreasing the probability that they will act as cyberbullies. Similarly, students with these emotional skills may have more resources to respond to cyberbullying situations, as well as in the face of anger or rage, which may lead to their being attacked via electronic means. A student who is able to understand and regulate their emotions may use these skills to modify negative emotions and thus act in a peaceful and adjusted manner.

## 5. Conclusions

The goal of this study was to analyze the predictive capacity of aggressiveness (physical aggressiveness, verbal aggressiveness, anger, and hostility) and emotional intelligence (attention, understanding, and emotional regulation) with respect to being a victim, aggressor or victim–aggressor of cyberbullying in adolescent populations. The results of the study reveal a higher probability of being either a victim, aggressor or victim–aggressor as the level of physical aggressiveness and anger increases. Further, the data indicate that having a greater level of understanding and emotional regulation decreases the probability of participation in any of the roles involved in cyberbullying. Therefore, it is recommended that preventive and intervention education programs consider these variables. Detecting physical aggressiveness or anger early on in adolescence and promoting assertive behavior in response to antisocial behavior and strengthening understanding and emotional regulation in students may be key to effectively intervening in this educational and social issue that is an increasing threat in our classrooms.

## Figures and Tables

**Table 1 ijerph-16-05079-t001:** Logistic regression for the probability of being a victim, aggressor or victim–aggressor based on aggressiveness.

Total Sample	Factors	B	S.E.	Wald	*p*	OR	CI 95%
Victim	Physical Aggression	0.030	0.013	5.605	0.018	1.03	1.00–1.06
	Anger	−0.055	0.017	9.925	0.002	1.06	1.02–1.08
	Constant	1.27	0.277	20.922	0.000	3.55	
Aggressor	Physical Aggression	0.050	0.012	17.844	0.000	1.05	1.02–1.08
	Anger	−0.084	0.017	25.645	0.000	1.08	1.04–1.10
	Constant	0.674	0.254	7.029	0.008	1.96	
Victim/Aggressor	Physical Aggression	0.116	0.020	34.709	0.000	1.12	1.08–1.17
	Anger	−0.170	0.027	38.510	0.000	1.07	1.02–1.09
	Constant	0.527	0.372	2.006	0.157	1.69	
Sex							
Boys							
Aggressor	Physical Aggression	0.061	0.029	4.446	0.035	1.06	1.00–1.13
	Anger	−0.076	0.023	11.433	0.001	1.03	1.00–1.06
	Constant	0.885	0.325	7.409	0.006	2.422	
Girls							
Victim	Physical Aggression	0.072	0.018	16.298	0.000	1.07	1.04–1.11
	Verbal Aggression	−0.066	0.032	4.377	0.036	1.04	1.01–1.07
	Constant	1.408	0.352	15.978	0.000	4.089	
Aggressor	Anger	−0.082	0.023	12.748	0.000	1.02	1.00–1.05
	Constant	0.560	0.322	3.021	0.082	1.751	
Victim/Aggressor	Physical Aggression	0.070	0.017	17.287	0.000	1.07	1.04–1.11
	Anger	−0.077	0.021	13.314	0.000	1.03	0.99–1.05
	Constant	1.183	0.363	10.624	0.001	3.263	
Grade							
E.S.O. (12–16 years)							
Aggressor	Physical Aggression	0.036	0.014	6.641	0.010	1.04	1.01–1.07
	Anger	0.057	0.026	4.974	0.026	1.06	1.01–1.11
	Constant	1.098	0.281	15.288	0.000	2.998	
Baccalaureate (16–18 years)							
Aggressor	Physical Aggression	0.071	0.020	13.004	0.000	1.07	1.03–1.12
	Anger	−0.082	0.025	10.503	0.001	1.02	1.00–1.03
	Constant	0.030	0.402	0.006	0.940	1.031	

B = coefficient; S.E. = standard error; *p* = probability; OR = odds ratio; CI = confidence interval at 95%.

**Table 2 ijerph-16-05079-t002:** Logistic regression for the probability of being a victim, aggressor or victim–aggressor based on emotional intelligence (EI).

Total Sample	Factors	B	S.E.	Wald	*p*	OR	CI 95%
Victim	Understanding	−0.060	0.013	22.056	0.000	0.94	0.92–0.97
	Repair	−0.029	0.012	5.426	0.020	0.97	0.95–0.99
	Constant	3.09	0.27	131.68	0.000	21.96	
Aggressor	Understanding	−0.053	0.012	19.019	0.000	0.94	0.92–0.97
	Repair	−0.050	0.012	18.161	0.000	0.95	0.93–0.97
	Constant	2.68	0.248	116.817	0.000	14.65	
Victim/Aggressor	Understanding	−0.109	0.022	24.586	0.000	0.89	0.86–0.94
	Repair	−0.097	0.020	22.315	0.000	0.90	0.87–0.95
	Constant	5.56	0.520	114.157	0.000	258.82	
Sex							
Boys							
Victim	Understanding	−0.047	0.017	7.587	0.006	0.95	0.92–0.99
	Repair	−0.043	0.017	6.064	0.014	0.96	0.93–0.99
	Constant	3.046	0.349	75.989	0.000	21.032	
Aggressor	Understanding	−0.042	0.016	6.712	0.010	0.96	0.93–0.99
	Repair	−0.052	0.016	10.136	0.001	0.95	0.92–0.98
	Constant	2.555	0.322	63.055	0.000	12.872	
Victim/Aggressor	Understanding	−0.068	0.012	31.308	0.000	0.93	0.91–0.96
	Constant	2.613	0.321	66.101	0.000	13.635	
Girls							
Victim	Understanding	−0.059	0.015	16.126	0.000	0.94	0.92–0.97
	Repair	−0.043	0.014	9.534	0.002	0.96	0.93–0.98
	Constant	3.548	0.346	105.262	0.000	34.737	
Aggressor	Attention	−0.025	0.012	4.110	0.043	0.98	0.95–0.99
	Understanding	−0.052	0.015	12.582	0.000	0.95	0.92–0.97
	Repair	−0.046	0.013	12.602	0.000	0.96	0.93–0.98
	Constant	3.128	0.335	87.326	0.000	22.839	
Victim/Aggressor	Understanding	−0.052	0.015	11.805	0.001	0.95	0.92–0.98
	Repair	−0.044	0.014	9.689	0.002	0.96	0.93–0.98
	Constant	3.534	0.353	100.341	0.000	34.245	
Grade							
E.S.O. (12–16 years)							
Aggressor	Understanding	−0.040	0.013	9.929	0.002	0.96	0.94–0.99
	Repair	−0.067	0.012	28.992	0.000	0.94	0.91–0.96
	Constant	2.885	0.266	117.651	0.000	17.898	
Victim/Aggressor	Attention	0.026	0.013	4.039	0.044	1.03	1.00–1.05
	Understanding	−0.056	0.015	15.003	0.000	0.95	0.92–0.97
	Repair	−0.048	0.014	12.125	0.000	0.96	0.93–0.98
	Constant	3.000	0.317	89.636	0.000	20.081	
Baccalaureate (16–18 years)							
Victim	Understanding	−0.085	0.016	29.585	0.000	0.92	0.89–0.95
	Constant	3.055	0.420	52.944	0.000	21.220	
Aggressor	Understanding	−0.090	0.015	37.377	0.000	0.91	0.89–0.94
	Constant	2.145	0.367	34.067	0.000	8.541	
Victim/Aggressor	Understanding	−0.071	0.016	20.823	0.000	0.93	0.90–0.96
	Constant	2.843	0.420	45.770	0.000	17.161	

B = coefficient; S.E. = standard error; *p* = probability; OR = odds ratio; CI = confidence interval at 95%.

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
