# Peer review of "Cyberbullying, Aggressiveness, and Emotional Intelligence in Adolescence"

_ijerph, 2019, doi:10.3390/ijerph16245079_

Round 1
Reviewer 1 Report
The study aims at investigating the predictive role of individual variables such as aggressiveness and Emotional intellingence, potentially adding some information to the great number of studies which investigated the predictors of cyberbullying. In this sense, I don't find striking elements of interest, rather the study could brings a confirmation to the existing literature, since its results rely on a large sample. However, my greater concern, is the treatment of the two precictors as separate constructs, both at the level of the hypothesis and at the level of the statistical analysis. Literature, in fact, provides evidence that aggressiveness and Emotional regulation, for instance, can be strictly intertwined, and a more complex model would account for this relation. As it is, all the considered predictors seems to have the same predicitve power both for cyberbullying and for cybervictimization, while this could not be the case if you would consider Aggressiveness as a predictor, EI as a mediator and Cyberbullying and Cybervictimization as outcome variables, in a Structural Equation Model. I would suggest, in this case, to include also Gender and age into the model, because some effects could be covered by the covariance.
Following this line, I would appreciate in the introduction a deeper discusiosn about the relation among the predictors and/or their differences due to age and gender. I would also suggest to clarify which is the theoretical reason behind the choice of the two predictors, and how they could interact in predicting differently Cybervictimization and Cyberbullying. Please provide also some description of the constructs of EI, explaining the theory behing the construct you have adopted.
The research question is quite weak, and it should be improved, adding more complexity to the study, as indicated above.
In the results section, beside the SEM suggested, I would appreciate a descriptive table with the frequencies of the different roles (cyberbully, cybervictims, cyber bully-victims, not involved) because it is not clear the number of subjects entered into the Logistic regression. In the SEM model you could measure the outcome variables according tho the scales of cyberbullying and cybervictimization, avoiding the dichotomization that, as it is now, it is not very clear.
If you decide to stay however only with Logistic regression and dichotomic variables, please control for Gender and Age and check for not involved students as well.
In conclusion, the large number of subjects and the variables you have considered could result in a interesting study, but as it is now, it does not reach its full potential.
Reviewer 2 Report
The topic under study is relevant and current. The predictive value of aggressiveness and emotional intelligence regarding cyberbullying is analyzed. However, there are a number of minor issues that have to be taken into account:
Page 1, line 36. The word "Recently" has to be deleted since the study referred to is from 2014 (five years ago).
Page 3, line 114. Not only the predictive capacity of emotional intelligence is evaluated, but also of aggressiveness. Specify it.
Pages 5 and 6. What is the reason why the verbal aggression dimension is not included in Table 1 or emotional attention in Table 2?
References section. Doi must be included in references numbered 31, 34 and 41.
References section. The journal name of the reference numbered 48 must be written in italics.
Reviewer 3 Report
In the abstract (lines 20-22), the description about the influence of Emotional Intelligence in Ciberbullying is not so clear.
In Discusion (lines 305-307), this sentence is not so clear, please, consider a review. Problably, the word "their" is incorrect.
Round 2
Reviewer 1 Report
I appreciate the improvement in the manuscript, altho my greater concern on the covariance between aggression and emotional intelligence has not been solved, since the authors have decided to keep separate Logistic Regression Models for the two predictors consideredwithout a clear motivation for this. Please insert a rationale in the introduction explaining your choice for not consider the relation between Aggressiveness and EI in cyberbullying but keeping them separate.
Age and gender in my opinion add relevance to the results, but do not solve fully the problem, which is theoretical and methodological at the same time.
I would appreciate e also to have an idea of how many subjects were classified into the dichotomic categories of roles: victims, bullies and victim-aggressors. Please add the descriptive table as requested.
However please check carefully the insertion of the new parts because they may be not inserted in the right position and/or they appear confusive in some parts (i.e. lines 113-116; and in the abstract).
